# Enzymes Immobilized into Starch- and Gelatin-Based Hydrogels: Properties and Application in Inhibition Assay

**DOI:** 10.3390/mi14122217

**Published:** 2023-12-08

**Authors:** Elena N. Esimbekova, Irina G. Torgashina, Elena V. Nemtseva, Valentina A. Kratasyuk

**Affiliations:** 1Institute of Fundamental Biology and Biotechnology, Siberian Federal University, 660041 Krasnoyarsk, Russia; esimbekova@yandex.ru (E.N.E.); torira@mail.ru (I.G.T.); enemtseva@sfu-kras.ru (E.V.N.); 2Laboratory of Photobiology, Institute of Biophysics of Siberian Branch of Russian Academy of Science, 660036 Krasnoyarsk, Russia

**Keywords:** starch, gelatin, enzyme immobilization, enzyme-based assays, inhibition assay, biosensors

## Abstract

The present work is a review of the research on using hydrogels based on natural biodegradable polymers, starch, and gelatin for enzyme immobilization. This review addresses the main properties of starch and gelatin that make them promising materials in biotechnology for producing enzyme preparations stable during use and storage and insensitive to chemical and physical impacts. The authors summarize their achievements in developing the preparations of enzymes immobilized in starch and gelatin gels and assess their activity, stability, and sensitivity for use as biorecognition elements of enzyme inhibition-based biosensors.

## 1. Introduction

Enzymes are traditionally used as biocatalysts in the food industry and in medical applications—both as the basis for drug synthesis and as direct therapeutic agents [1,2,3,4]. A relatively new area of application for enzymes is their use as biorecognition elements in biosensors [5,6,7,8]. Yet, as native enzymes are adapted for functioning in the cellular environment, their properties are often unsuitable for large-scale use and need to be modified [9]. Indeed, most enzymes occur in soluble form and, thus, are sensitive to changes in physical and chemical environmental factors (temperature, pH, ionic strength). The problem can be solved by immobilizing the soluble enzymes to produce their solid forms [10,11,12,13].

The original purpose of enzyme immobilization was to retain an enzyme (or even several enzymes) within a certain space. Since the 1960s, enzyme immobilization has been used as a means to extract and reuse enzymes and as a way to stabilize them. Although much has been achieved so far, there are still challenges faced by researchers, and the choice of the method for immobilizing each particular enzyme remains rather the state of the art than the mature discipline [14].

This is even more true for the so-called multi-enzyme immobilization. The co-immobilization of the functionally related enzymes is one of the methods of constructing multi-enzymatic cascade reactions, which ensures improved cascade enzymatic activity via substrate channeling and/or co-factor regeneration [15]. In some cases, multi-enzyme immobilization additionally resulted in enhanced enzyme stability and the ease of recovery for reuse [16].

Among the considerable variety of methods, one of the simplest ways to immobilize enzymes is to entrap biocatalysts in the polymer gel matrix. This method has been particularly popular for designing biosensors because the immobilized enzyme may be easily located on the tip of the sensors [17]. The advantage of this approach is the achievement of increased stability of the enzymes immobilized in the highly chemically, mechanically, and thermally stable polymer gels [18,19]. The important criteria for choosing the carrier for producing an immobilized biocatalyst with improved properties are its shape, size, processability, and physicochemical properties [13]. The main disadvantage of immobilizing enzymes into polymer gels is that the polymer matrix can hinder substrate diffusion to the enzyme, which may lead to a decrease in the catalytic efficiency of the immobilized preparation [20]. The carriers with porous structures are the most suitable for this purpose, as they have a larger surface area for enzyme immobilization and less significant diffusion limitations for substrates and inhibitors [13]. Examples of such carriers are synthetic and natural gels with gelation conditions that are compatible with the stability of the enzyme [13]. Interestingly, the unique functional properties of hydrogels as enzyme carriers are related to the gel-like properties of intracellular and extracellular environments, so the entrapment in hydrogels could be considered a bio-inspired and cell-mimicking technology [21,22].

Among the wide variety of polymers used for enzyme immobilization, natural biodegradable polymers deserve special attention [11,19]. Increasing environmental awareness encourages the search for natural biodegradable polymer materials suitable for enzyme immobilization [23]. A characteristic property of natural biodegradable polymers such as agarose, starch, alginate, chitosan, and gelatin is that their gelation occurs under physiological conditions, and thus, the immobilized enzymes retain their functions [13,24]. Hydrogels based on these natural polymers have been used to entrap enzymes. For instance, tyrosinase was extracted from the plant source *Amorphophallus companulatus* and immobilized in a composite of two biopolymers: agarose and guar gum. This electrochemical enzyme electrode for dopamine had a reusability of up to 15 cycles and a shelf life of more than 2 months [25]. Lipase immobilized on polysaccharides chitin and chitosan showed reasonably high activity and stability [26]. Horseradish peroxidase and lactate oxidase were successfully immobilized within alginate hydrogels, and the proposed biosensors showed good stability [27].

Because of their natural origin, wide availability, and low cost, starch and gelatin polymers are the most commonly used natural polymers in the food industry, pharmaceutics, medicine, and biotechnology [28,29,30,31,32]. The unique physicochemical properties of starch and gelatin and their biodegradability and biocompatibility under physiological conditions are the reasons for the increasing interest of researchers in these polymers [32,33,34,35].

The present work is a review of our studies on using natural biodegradable hydrogels based on starch and gelatin as carriers for immobilizing enzymes.
Section 2
of this review gives a detailed description of the structure of the starch and gelatin polymers, the mechanisms of gelation of these polymers, and their properties essential for enzyme immobilization.
Section 3
presents data on the properties of the enzymes immobilized in starch and gelatin gels. The possibility of stabilization of enzymes by immobilizing them into starch and gelatin gels is discussed in detail, and the mechanisms of improving the thermostability of the enzymes and their interactions with inhibitors are considered.

## 2. Molecular Structure of the Starch and Gelatin Biopolymers and Architecture of Their Hydrogels

### 2.1. Starch—A Storage Polysaccharide of Plants

Starch is known to be the most abundant organic compound in nature after cellulose and the main storage form of carbohydrates [36,37]. Starch is accumulated in cereals, legumes, roots, tubers, and some fruits in the form of insoluble, semi-crystalline granules of various sizes (diameter of 1–100 µm) and shapes (polygonal, spherical, lenticular, etc.) [32,37]. The composition and structure of starch granules may vary between plant species, cultivars of the same species, and even within the same plant cultivar grown under different conditions [38,39]. It is believed that such variability is associated with the complex nature of starch biosynthesis.

The main components of starch are two polymers of D-glucose. The first is amylose—an essentially unbranched α-(1–4) linked glucan with a molecular weight of 30–3200 kDa. The second is amylopectin—a polymer with a highly branched structure, which consists of chains of α-(1–4) linked glucose with numerous α-(1–6) branching links. The molecular weight of amylopectin varies within a wider range—from 10,000 to over 500,000 kDa. An important property, which largely determines the physicochemical characteristics of starch hydrogel, is the amylose-to-amylopectin ratio [39,40,41]. The majority of starches contain 15–35% of amylose.

The spatial organization of amylose and amylopectin within starch granules is rather complex. Amylose is found naturally in three crystalline modifications, which results in different packing densities. Its linear molecule can contain helical segments of three types, depending on the degree of hydration. Starch granules contain both crystalline and amorphous regions in alternating layers. Many small crystalline areas are formed by clustered branches of amylopectin packed together in double-helical folding, while amylose molecules are located between the amylopectin molecules. The detailed description can be found elsewhere [41,42]. Thus, in spite of being formed by the same monomer in similar linking, the starches from different plant species are composed of polymers of different molecular weights and variously packaged into granules, which underlies the variability of the properties of starch hydrogels.

The interaction of starch with water depends on the temperature. Starch granules are insoluble in cold water, but under heating, they undergo swelling, pasting, and gelatinization. Upon the continued heating and hydration of the starch granules, the swelling begins in the amorphous, inter-crystalline regions. Then, the dissociation of double-helical regions takes place, and the amylopectin crystallite structure disappears, which is accompanied by an increase in viscosity. Starch gelatinization is generally defined as the irreversible disruption of molecular orders resulting in starch solubilization. Many factors determine the temperature of initial gelatinization and the range over which it occurs: the starch concentration, granular type, heterogeneities within the granule population, and others [42]. The gelatinization temperature of most starches is between 60 and 80 °C [43].

Starch is used in its native or modified form. Native starch is a convenient biopolymer and copolymer in biomedical and pharmaceutical applications due to its renewability, biocompatibility, biodegradability, and relative cheapness. However, to overcome shortcomings of this material such as poor mechanical properties, high hydrophilicity, and others [23,32], various modification techniques are used, both chemical (substitution, cross-linking, oxidation, esterification, etc.) and physical (grinding, extrusion, plasticization, etc.) [23,42].

### 2.2. Gelatin—A Structural Polypeptide of Mammals

Gelatin is a water-soluble biopolymer of the polypeptide type. Just as starch is thought to be one of the most abundant polysaccharides in plants, gelatin can be said to be one of the most abundant proteins in mammals [29,44]. In the form of a triple helix, it makes up mechanically strong collagen fibers in tendons, bones, skin, cornea, and other connective tissues [44]. The basic unit of collagen is supposed to be a chain with a molecular weight of about 95 kDa, consisting of a repeating triplet of the Gly-X-Y amino acids. In this triplet, X is often proline, and Y is sometimes hydroxyproline. These chains can be covalently bound in twos, threes, or more, which, along with the formation of triple helices, is the cause of the unique mechanical properties of collagen. Gelatin chains are extracted from natural collagens using the methods of acidic/alkaline hydrolysis and thermal/enzymatic treatment. The preparation method affects some properties of the resulting gelatin, such as an isoelectric point, average molecular weight, and others. Unlike other biopolymers or synthetic hydrophilic polymers, gelatin has both acidic and basic functional groups, forms a specific triple-stranded helical structure, which is absent in most synthetic polymers at low temperatures, and specifically interacts with water [45].

Gelatin hydrogels are formed after the dissolution of gelatin in warm water (≥40 °C) and subsequent cooling at the biopolymer concentration above some critical value (which is typically 0.4–1%). The transition to the gel state is accompanied by the formation of physical crosslinks or junction zones due to the ordering of some segments of the neighboring polypeptide chains into triple-helical collagen-like structures [46].

Along with advantages such as availability and biocompatibility, gelatin demonstrates good foaming, emulsifying, film-forming, and gelling properties, which contributes to its wide application in technical, medical, food, pharmaceutical, cosmetic, and other industries [45,47,48,49,50]. The large variety of applications is also made possible by the availability of the multiple functional groups of gelatin molecules, allowing for the cross-linking of polymer structures, their coupling with targeted ligands, and other modifications useful in a particular application [48,49,51]. In the vast majority of cases, it is modified gelatin gels rather than native ones that are used as carriers of bioactive components.

### 2.3. Properties of the Starch and Gelatin Hydrogels

The ability of widely available biopolymers such as starch and gelatin to form hydrogels and the properties of their hydrogels are determined by the chemical structure of the constituent macromolecules. Because of the diverse composition of natural sources of the biopolymers and the employment of various extraction techniques, hydrogels based on them differ in their physicochemical properties.

Although the macromolecules of gelatin and starch differ in their nature, the gels formed by them have some similar properties. They both are physical gels, i.e., their three-dimensional crosslinked network is stabilized by noncovalent interactions (hydrogen bonds, helix formation, complexation at the network junction points). The helix formation is the main mechanism of the crosslinking both for polypeptide chains of gelatin and for polysaccharides of starch. For the formation of the hydrogel network by both biopolymers, the heating in water and subsequent cooling are needed.

However, there are obviously more differences than similarities between starch-based and gelatin-based hydrogels (Figure 1). These biopolymers have different critical concentrations for gel formation. The gelatin solution is known to form a gel when the polymer concentration is greater than 0.4–1.0% [46], whereas the starch solution under the same conditions becomes only a viscoelastic paste. The paste transforms into the gel at the starch concentration >6% [52]. Gelatin- and starch-based gels are considered to be of different types—“strong” and “weak”, respectively, with essentially different rheological properties. Both strong and weak gels respond as solids at small deformations, but while the former are also solids at a larger deformation, the latter flow under such conditions [46]. The gelatin gels contain extended cross-links or junction zones formed by a partial reversion to ordered triple-helical collagen-like sequences. The starch gel is a result of the network formation by interactions between leached amylose molecules, while the more numerous amylopectin molecules (having lost their crystallinity) mostly remain in a random coil state. Such distinctions led to different mesh sizes of these gels. For the gel of 5% gelatin, the mesh size was estimated at about 51 Å [53]. The starch gel matrix was found to have a much greater mesh size, 35–36 nm [54], the mesh being formed by amylose chains, while amylopectin functioned as a solute in water that was compartmentalized by the network [55].

In our previous work [56], we studied a correlation between the rheological properties in macro- and micro-scales of gelatin- and starch-based solutions used for the immobilization of the enzymes. Opposite trends were observed for the relationship between microviscosity η_m_ and macroviscosity η for the two biopolymers: η_m_ << η for gelatin and η_m_ >> η for starch solutions. The temperature dependence of η_m_ followed the monoexponential decay law in all samples over the whole temperature range, 15–50 °C, indicating the insensitivity of microviscosity to gel mesh melting under heating.

Thus, gelatin-based gels are rugged, resembling solids in their physical state, forming some kind of a skeleton through which small molecules can easily diffuse. In contrast, a suspension of starch, especially potato starch, which is characterized by high amylopectin content, is not able to form strong gels, but it can provide a high degree of immobilization, probably due to hydrophobic interactions.

## 3. Entrapment of the Enzymes within a Porous Matrix of Starch and Gelatin to Use in Inhibition Assay

### 3.1. Methods of Producing Preparations Based on Enzymes Immobilized in Starch and Gelatin Gels

Over the past few decades, researchers have gained considerable experience in developing various procedures for immobilizing cells, individual enzymes, and multi-enzyme systems in starch and gelatin gels.

The basic procedure of immobilizing enzymes in gels is very simple. It consists of just a few major steps: the preparation of the gel and the mixing of the gel with the enzyme solution. The final step may include pipetting the mixture onto the support and drying (Figure 2). This is the way to produce single-use reagents containing the preset amount of enzyme (Figure 3) [57,58,59,60]. In this form, they can be used in cuvette-based instruments (spectrophotometer, luminometer). To record the activity of the immobilized enzyme, the dry disk is placed into the tested solution, and a required component (usually one of the substrates) is added to start the enzymatic reaction.

The above-mentioned method of immobilizing enzymes by entrapping them in starch and gelatin hydrogels without any previous modifications was successfully used to produce stable preparations of butyrylcholinesterase (BChE), trypsin (Try), firefly luciferase (FLuc), and a bioluminescent bi-enzyme system of luminous bacteria: NAD(P)H:FMN-oxidoreductase and luciferase (Red + Luc) [58,59,60].

In the vast majority of cases, the immobilization of enzymes into the starch and gelatin gels is performed by using various mixtures or composites rather than individual polymers. For example, for co-immobilizing lipase and protease, starch was combined with chitin [61]. Dialdehyde starch was used as a coupling agent to prepare a chitosan carrier to immobilize the xylanase from Aspergillus niger [62]. Also, dialdehyde starch with nanocrystallization was used as a cross-linking agent to immobilize Candida Antarctica lipase B on a carrier with Fe_3_O_4_ as the core [63]. The same is true of gelatin. For instance, arginase and urease were immobilized on the surface of the pH electrode by using a gelatin membrane, which was then cross-linked with glutaraldehyde [64]; laccase and catalase were immobilized using a mixture of gelatin and alginate [65,66,67]. A comprehensive review of gelatin-based drug delivery systems [49] shows that among the >60 methods of production of particles for the delivery of bioactive compounds, only 14 use gelatins without any crosslinking procedure.

There are literature data on another application of enzymes immobilized in gel, particularly their multiple uses in analyses. For example, β-galactosidase was immobilized on a biocomposite of alginate and gelatin crosslinked with genipin, and it maintained 90% relative activity after 11 reuses in a batch process of lactose hydrolysis [68]. The electrode based on glucose oxidase immobilized onto gelatin by cross-linking with chromium III acetate retained more than 60% of its activity after 20 repeated uses [69]. Urease purified from pigeonpea seeds was immobilized on gelatin beads via cross-linking with glutaraldehyde. These beads can be reused more than 30 times without much loss of enzyme activity [70]. A major disadvantage of this approach is the necessity to wash the immobilized enzymes after each use, which certainly makes the procedure more complicated. However, the advantage of this method is that it saves on expensive enzyme preparations.

### 3.2. Characterization of Preparations Produced by Immobilizing Enzymes in Starch and Gelatin Gels

High enzyme activities are achieved by individually selecting immobilization conditions, preparation composition, and/or the method of using the immobilized enzyme preparation. An increase in the enzyme content of the preparation clearly leads to an increase in the recorded activity of the immobilized enzyme. The efficiency of enzyme immobilization as dependent on the conditions of production of the immobilized preparations (such as drying temperature, enzyme and/or stabilizer content, and the buffer employed) is estimated using the immobilization yield parameter—the ratio of the maximal activity of the immobilized enzyme (under optimal conditions) to the activity of the same amount of the free enzyme.

The functional properties of immobilized enzymes are largely determined by the source of starch or gelatin. The study [71] demonstrated that the best carrier for immobilization of the Red + Luc bi-enzymatic system of luminous bacteria is potato starch gel. The probable reason is that starch produced from different natural sources contains dissimilar amounts of amylose and amylopectin (see Section 2), which, in turn, influences the starch gelatinization process [39,40,72].

Sometimes, starch is modified to change its characteristics and achieve high mechanical properties, thermal stability, and acid stability; enhance gel clarity, film formation, and adhesiveness; and reduce retrogradation [73,74,75]. For example, maltogenic α-amylase was used to modify granular waxy, normal, and high-amylose maize starches for improved functional attributes. This modification reduced the retrogradation rates of granular waxy and normal maize starches and decreased their pasting viscosity [76]. A similar result was obtained for faba bean, lentil, sweet potato, and pea starches modified by maltogenic amylase [77,78]. In another study, the shortening of amylose and amylopectin chain lengths for reducing retrogradation rates was achieved by the sequential addition of glucan transferases, β-amylase, and transglucosidase [79]. There are detailed studies describing methods of dual modification of starch, including dual chemical modification, dual physical modification, dual enzymatic modification, and dual heterogeneous modification. As a rule, they are used to improve various properties of starch in the food industry [80]. However, starch modification often leads to a significant added value in the final product [73].

Similarly, the properties of gelatin as a carrier for enzyme immobilization are determined by both its source and gel preparation technique. Being brittle, gelatin is seldom used alone, without being mixed with other substances, in immobilization procedures. To tune the mechanical strength and bioactivity, gelatin can be easily modified via chemical reactions and physical methods. The major ways to obtain its various derivatives are crosslinking, grafting, or blending [81].

In a number of studies, enzymes with enhanced properties were successfully produced by using gelatin composites with polysaccharides such as alginate [82,83]. For example, an alginate–gelatin hydrogel matrix was synthesized and used as immobilization support for *Mucor racemosus* lipase. The immobilized lipase catalyzed olive oil hydrolysis; ultrasound examination showed a significant increase in hydrolysis rate compared to free lipase [82]. A commercially available inulinase from *Aspergillus niger* was immobilized in calcium alginate–gelatin capsules and the immobilization yield of 86% was demonstrated [83].

An important objective is to select the polymer concentration of the gel at which the gel will meet the following requirements: (1) precise pipetting of microliters of the gel onto the support; (2) sufficient mechanical strength of the preparations after drying; (3) rapid rehydration; (4) a minimal effect on the rate of enzymatic reaction (via molecular interactions with the enzyme or substrates or through diffusion limitations).

For instance, 3% starch gel was chosen as a carrier for producing single-use BChE preparations [84]. It was also found that BChE activity was not affected by variations in potato starch concentration in the gel within a 1–4.2% range (Figure 4). However, at low starch concentrations, the preparations were too fragile, while the use of high starch concentrations caused difficulties in both the fabrication of the preparations (pipetting complications) and their use. The rehydration of such preparations (soaking in buffer solution) took a long time, thus increasing the assay duration. For similar reasons, in experiments with BChE immobilization in gelatin gel, 1.4% of gelatin was chosen as the optimal percentage [58,84].

Table 1 lists values of immobilization yields of some enzymes immobilized by entrapment within the porous matrices of starch and gelatin. For the Red + Luc bi-enzymatic system, immobilization in the starch gel results in a greater value of immobilization yield. The immobilization of the bi-enzymatic system in the gelatin gel leads to a considerable loss of enzyme activity. The likely reason for this is that gelatin is able to sorb charged biomolecules, including proteins, via polyionic complexation [33]. Moreover, protein–protein interactions may occur between gelatin and enzymes, leading to a partial change in enzyme conformation.

In contrast, higher BChE activity (comparable to the activity of the free enzyme) was achieved by the entrapment of the enzyme in the gelatin gel (Table 1). The starch gel was completely unsuitable for immobilizing FLuc [59]. A possible reason for that was that components of the buffer solution used for this reaction interacted with starch gel, preventing the disks from drying completely even in the long term. Those disks could not be detached from the hydrophobic film for activity assay. In the gelatin gel, though, the function of the firefly luciferin–luciferase system did not change substantially even when polymer content reached 5% [86]. By immobilizing FLuc in the 1% gelatin gel, we produced high-activity single-use preparations for assessing microbial contamination based on ATP content [59,87]. Although other authors also reported successful immobilization of enzymes in gelatin gel, an essential difference was the use of additional components, primarily alginate. The activity yield of bovine liver catalase immobilized in the gelatin-alginate hydrogel was 92% [67].

Another important aspect that influences the choice of the carrier or concentration of the gel for immobilization is the possible effect of diffusion processes on the activity of the enzymes entrapped within a porous matrix. The decrease in the activity of immobilized enzymes may be caused by diffusion limitations if the substrate diffusion rate is lower than the rate of substrate consumption by the enzyme [88,89,90,91]. A study of the microviscosity of gels based on gelatin and starch used for immobilization by employing a fluorescent molecular rotor probe showed that starch can create more obstacles to the diffusion of low-molecular-weight substances than gelatin [56]. The explanation for this lies in the structure of the gels produced by these biopolymers. The gelatin gel contains extended physical cross-links formed by triple-helical collagen-like structures. The starch gel is a result of the network formation by leached amylose molecules, whereas more numerous amylopectin molecules (that have lost their crystallinity) mainly stay in the random coil form.

A way to minimize the diffusion of intermediates among the enzymes and increase their overall activity is to design multi-enzyme systems. In that case, the product of one enzyme can be channeled to act as a substrate for the second enzyme. The obtained multi-enzyme system may enhance the overall activity based on high local substrate concentrations [15]. In our research, this idea was implemented by the co-immobilization of the bi-enzyme system of luminous bacteria Red + Luc. Moreover, the simple procedure of immobilizing enzymes into the starch and gelatin gels and the high activities of the immobilized enzymes provided the basis for producing multicomponent enzymatic preparations that contained not only different enzymes but also substrates or indicators of enzymatic activity [57,58,59,85].

The multicomponent preparations contained various combinations of co-immobilized enzymes Red and Luc and substrates tetradecanal, NADH, and FMN [71]. The preparations containing Red + Luc + tetradecanal or Red + Luc + tetradecanal + NADH co-immobilized into starch gel exhibited the highest immobilization yield (of about 100%). The multicomponent preparations, including other combinations of co-immobilized enzymes and substrates (such as Red + Luc + FMN; Red + Luc + NADH; Red + Luc + FMN + tetradecanal or Red + Luc + FMN+NADH), showed relatively low levels of the recovery of activity (from 1 to 5%). Again, 3% potato starch gel was the best carrier for producing the preparations (Figure 5) [71]. The multicomponent preparations, including the Red + Luc enzymes and the tetradecanal + NADH substrates, were trademarked Enzymolum, and the fabrication process and the inhibition assay procedure were patented [92,93]. The process of producing multicomponent preparations of the bi-enzymatic Red + Luc system co-immobilized with substrates into starch gel served as the basis for fabricating single-use microfluidic chips. The chips used in combination with a specially designed portable bioluminometer enable rapid screening of a large number of samples [94,95].

Another multicomponent preparation contained FLuc and its substrate, D-luciferin, co-immobilized into 1% gelatin gel [59]. The immobilization yield of co-immobilized FLuc and D-luciferin was 50%. It was sufficiently high for the multicomponent preparation to be used in microbial contamination assay. One more example of the successful co-immobilization of the components of the analytical enzyme system is a multicomponent preparation based on BChE co-immobilized with the thiol group indicator—5,5′-dithiobis(2-nitrobenzoic acid) [85,96].

Such multicomponent immobilized enzyme preparations can be used as reagents for dry chemistry not only in routine assays but also as biological modules in enzymatic biosensors [57].

Thus, immobilized enzyme preparations with the necessary catalytic activity but different properties, which meet the requirements of a specific analytical problem, can be produced by varying the conditions of immobilization and composition of the multicomponent preparation (by selecting the polymer hydrogel and its concentration, by incorporating substrates/indicators).

### 3.3. The Effect of the Gel Matrix on Stability of the Enzymes Immobilized in the Starch and Gelatin Gels

The main objective of the immobilization of enzymes is to enhance their stability during storage and usage under different conditions. The choice of the carrier determines the microenvironment of the enzymes, in which they should remain stable when exposed to various physical and chemical environmental factors, e.g., retain their high activity in a wide temperature or pH range in solutions with different ionic strengths, etc.

The most extensive data have been obtained on the effects of starch and gelatin hydrogels on the kinetic properties of the bi-enzymatic system of luminous bacteria Red + Luc. The study [97] demonstrated an increase in the apparent Michaelis constants (Km app) in the immobilized Red + Luc systems for all three substrates compared with the free enzymes (Table 2), which can be explained by the limitation of the conformational lability of protein molecules reflected in the rate of the catalytic activity. Another reason for the increase in the Michaelis constants may be complications with the delivery of the substrates to the enzymes immobilized in the hydrogels [97].

The reason for the diffusion limitations, which are the common drawback of immobilized enzymes, is that for effective enzymatic catalysis, the substrates should reach the active site of the immobilized enzyme through the reaction medium, and the products should diffuse away from it. To overcome this drawback, a number of strategies are employed, including thermally reversible hydrogels and carriers, the immobilization of enzymes on the surface of the carrier, and others [98]. The co-immobilization of the coupled enzymes and their substrate is also a helpful measure to minimize diffusion limitations.

One of the definite proofs of the successful immobilization of enzymes is their activity retained during long-term storage and, preferably, in a wide temperature range.

Immobilized in starch gel, the Red + Luc enzymes retained their activity for two years versus about one month in gelatin gel. The activity of the free Red + Luc enzymes in solution, however, dropped to zero in three days, even when it was stored at 8 °C [71]. Trypsin immobilized in starch gel stored at 8 °C for 4 months retained 80% of its activity [60]. The most impressive results were obtained for BChE immobilized (alone or together with the thiol group indicator—5,5′-dithiobis(2-nitrobenzoic acid)) in both starch gel and gelatin gel. The preparations retained high activity for more than 10 months at 8 °C [58].

Storage stability appears to be achieved thanks to the removal of free water from the starch and gelatin gels during drying of the gel/enzyme mixture. Thus, the movements of molecules in the polymer matrix are nearly completely quenched; hence, the structural rigidity of enzyme molecules is increased [57]. As a result, enzyme molecules are fixed in a polymer network, which prevents their denaturation and inactivation during storage.

Thermal stability is one of the most important parameters of the enzymes employed in analytical studies. It is used to estimate the degree of stability of enzymes’ catalytic activity in response to the effects of temperature and, hence, to determine both the conditions for producing immobilized preparations and the conditions of their use.

Immobilization of Red + Luc in starch and gelatin gels widened the temperature range for maintaining the function of the immobilized enzymes (Table 2) [97]. The optimal temperatures for the enzymes immobilized in the starch and gelatin gels were 5–50 °C and 15–50 °C, respectively, while the free enzymes were completely inactivated at 40 °C. The starch and gelatin gels produced a stabilizing effect on BChE as well [84].

Several studies were devoted to the mechanisms of the thermal inactivation of enzymes in starch and gelatin hydrogels [84,99,100]. The oligomeric enzymes (such as BChE and Red) functioning in hydrogels were found to be thermally inactivated through the dissociative mechanism. The kinetic curves in the first-order coordinates have kinks suggesting different rates of at least two sequential processes: dissociation into monomers, which occurs at a faster rate, and denaturation of the monomers, which occurs at a slower rate. The kinetic curves of the thermal inactivation of BChE in gelatin and starch solutions at different temperatures are shown in Figure 6 as examples [101]. It is important that the same mechanisms of thermal inactivation operate in all studied media (buffer water solution, starch, and gelatin gels). There are data indicating an increase in the activation energy of the thermal inactivation of BChE in starch and gelatin hydrogels [84].

Literature data suggest that the activity of immobilized enzymes is retained under longer exposure to high temperatures compared to the free enzymes, indicating enhanced thermal stability of enzymes in the starch and gelatin environments [84,99]. However, the effect of thermal stabilization by starch and gelatin gels is not observed for all enzymes. As reported in [100], the calculated values of activation energies of the thermal inactivation process in the stage of dissociation of *Vibrio fischeri* NAD(P)H:FMN-oxidoreductase did not significantly differ from the corresponding parameter in the buffer solution.

The profiles of pH dependencies of Red + Luc activity also changed in the starch and gelatin gels (Figure 7a) [102]. However, the pH optimum of Red + Luc (6.7–6.9) did not change when starch was added to the reaction mixture; when gelatin was added, the pH optimum shifted to a more alkaline value, pH 7.2. In addition, gelatin and starch had a stabilizing effect on Red + Luc at alkaline pH; as a result, the acceptable pH range became 6.8–8.1.

The immobilized enzyme preparations in the form of dried disks remained active at pH values varied within an even wider range [97]. For example, Red + Luc immobilized into starch gel and dried retained 100% activity at pH values between 5.8 and 8. The pH optimum of the bi-enzymatic Red + Luc system immobilized into gelatin gel had a narrower range: 6.6–7.3 (Table 2, Figure 7b).

The likely reason for such effects of pH on the activity of immobilized enzymes is that the charged groups of the carrier used for immobilization affect the distribution of protons in the microenvironment of the immobilized enzymes [97]. This effect is evident for the enzymes immobilized in the gelatin gel, which has charged groups. The pH dependence of the activity of the bi-enzymatic system immobilized in the starch gel is weak because of the electroneutrality of the carrier.

Similar results were demonstrated in the analysis of the dependence of bi-enzymatic system activity on the ionic strength of the solution. Both gelatin and starch had a stabilizing effect on Red + Luc at low ionic strength, and it was possible to use smaller amounts of buffer salts when the immobilizing reagents were prepared [102].

Thus, in most cases, starch and gelatin gels have a stabilizing effect on enzymes exposed to the physical and chemical denaturation factors.

### 3.4. Using Enzymes Immobilized in Starch and Gelatin Gels in Inhibition Assay

For enzymes to be used as biorecognition elements of enzyme inhibition-based biosensors, immobilization is a necessary step. The major difficulty is that while the aim of the stabilization of the enzyme is to make it insensitive to outer physicochemical factors, it must remain sensitive to the effects of the analytes. That is, two opposite objectives need to be achieved. Therefore, considerable research has been conducted on the effects of carriers on the sensitivity of immobilized enzymes to inhibitors.

At present, gelatin is a considerably more common material for the fabrication of biosensors than starch. It is extensively used to design biosensors for detecting analytes in medical diagnostics (glucose, hydrogen peroxide, urea, amino acids, pesticides), testing food products, and monitoring the environment [103,104,105,106]. The study [103] describes in detail the use of gelatin as a suitable matrix for the immobilization of biorecognition elements to ensure their high stability and durability. For example, cholinesterases immobilized in gelatin gel are widely used in biosensors for both semi-quantitative and integrated determination of inhibitors of these enzymes [107]. The arginase and urease enzymes were co-immobilized on the surface of the pH electrode by using a gelatin membrane cross-linked with glutaraldehyde [64].

Furthermore, the immobilization of enzymes in the starch and gelatin gels is a way to vary their sensitivity to inhibitors. The following methods can be used to increase the sensitivity of the reagent to pollutants: varying the composition of the immobilized reagent; varying the order of components added to the enzymatic system and test sample; varying the ratio of the reaction mixture and test sample; introducing an additional step of reagent incubation in the test sample solution [108].

The sensitivity of the enzyme to inhibitors has been best studied for the bi-enzymatic Red + Luc system co-immobilized into starch gel with its substrates. Comparing the sensitivity of the soluble and multi-component immobilized bi-enzymatic systems to phenols, quinones, and heavy metals shows that the sensitivity of the soluble bi-enzymatic Red + Luc system to the studied pollutants was higher compared to that of the immobilized enzymes, but the values of IC_50_ of the immobilized preparation were determined as comparable with or lower than their maximum permissible concentrations (MPC) [108]. Thus, the immobilized reagent enables measurements at the MPC level or even lower for almost all the studied substances, and it can be used for biotesting natural water and wastewater [108].

BChE immobilized in starch gel also demonstrated high sensitivity. BChE preparations based on starch gel showed a 74% reduction in activity in the 0.05 µM malathion solution, which corresponds to the MPC of malathion [85]. At the same time, preparations based on gelatin gel were less sensitive to organophosphorus compounds compared to the preparations based on starch gel. The activity of BChE immobilized in the gelatin carrier did not change in the presence of 0.5 µM of malathion—the amount is 10 times higher than the MPC of this pesticide. This could be caused by the high degree of stabilization of the enzyme by the gelatin carrier [85].

The sensitivity of trypsin immobilized in starch gel was assessed in experiments with pollutants and food additives [60]. The sensitivity of trypsin immobilized in starch gel to CuCl_2_, potassium sorbate, and CrCl_3_ was comparable to the sensitivity of the free trypsin.

The sensitivity of trypsin, BChE, and Red + Luc was largely determined by the time of incubation of the enzyme preparation in the inhibitor solution [60,85,108]. Additional incubation of the immobilized bi-enzymatic Red + Luc system in the copper sulfate solution for 5 min increased the sensitivity of the enzymes to that toxicant by a factor of 10 [108]. During the incubation of immobilized preparations in aqueous solutions of inhibitors, the starch was rehydrated, diffusion processes were accelerated, and the enzyme became more accessible to inhibitors and substrates. Thus, the sensitivity of the sensor based on enzymes immobilized in starch gel can be controlled by varying the time of incubation.

## 4. Conclusions

Natural polymer carriers for enzyme immobilization are certainly good candidates for wide use, but it is difficult to select a carrier that would meet all specific requirements. For instance, to achieve simultaneously high stability of the immobilized enzymes and their sensitivity to inhibitors is to reach two opposite objectives, and the only solution is to approach the optimum by selecting the carrier matrix experimentally. Yet, enzyme immobilization in starch and gelatin hydrogels followed by drying is the approach that has definite advantages: the immobilization process is simple to perform; enzymes retain high catalytic activity; enzymes are stable during storage and use; the sensitivity of the preparations to inhibitors can be varied. An added advantage of the enzymatic preparations based on starch and gelatin is that no special conditions are needed to store them. In contrast to the soluble enzyme forms, the enzymes immobilized in starch and gelatin gels can be used not only in the laboratory but also in the field.

Despite the many advantages of using natural hydrogels to immobilize enzymes, certain problems arise. For example, retrogradation or destruction of native starch changes the physicochemical properties of the gels, which may lead to changes in the properties of the enzymes immobilized in the starch gel. To avoid this, researchers suggest performing additional modifications of starch and gelatin, among which crosslinking is the most widely used procedure. Another likely problem is diffusion limitations. This problem can sometimes be overcome by conducting multi-enzyme immobilization.

Finally, based on the results of numerous experiments and taking into account the properties of starch and gelatin, such as biocompatibility, low cost, and availability, it seems safe to say that these natural polymers should be considered among the best carriers for enzyme immobilization.

## Figures and Tables

**Figure 1 micromachines-14-02217-f001:**
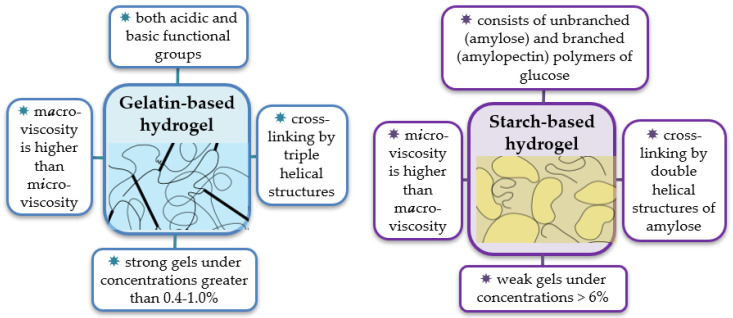
The schematic representation of the structure of the gelatin- (**left**) and starch-based (**right**) gels with indication of some distinctive properties.

**Figure 2 micromachines-14-02217-f002:**
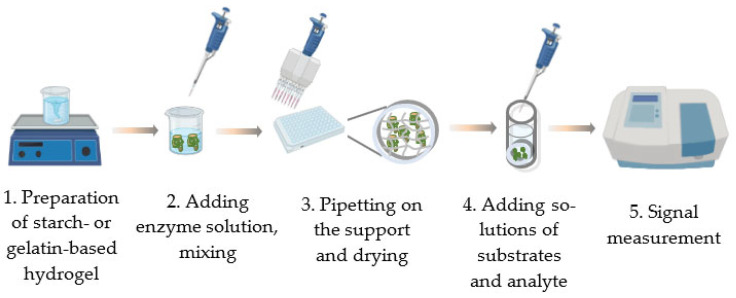
A schematic of fabricating and using enzyme preparations immobilized in starch or gelatin hydrogel.

**Figure 3 micromachines-14-02217-f003:**
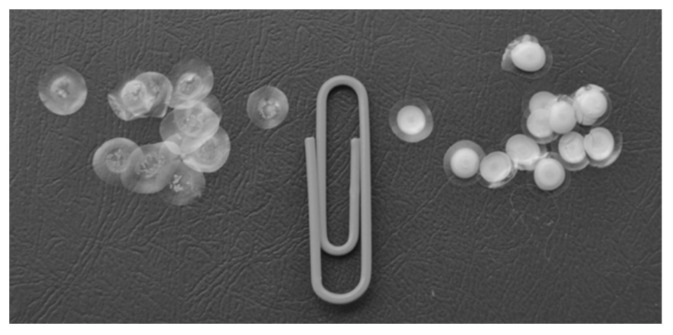
Exterior view of the multicomponent preparation based on the bi-enzymatic Red + Luc system (left—gelatin-based disks, right—starch-based disks). The gel-immobilized enzymes are dry disks 4–6 mm in diameter, 50–60 µm thick, weighing 9 ± 0.5 mg (DW). Reproduced from Ref. [57] with permission from Pleiades Publishing.

**Figure 4 micromachines-14-02217-f004:**
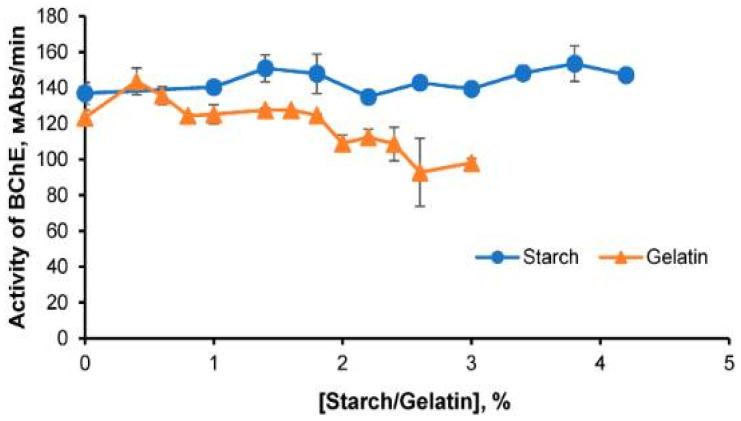
The rate of enzymatic hydrolysis of butyrylthiocholine iodide depending on the concentration of starch and gelatin. Reproduced from Ref. [65] with permission from MDPI.

**Figure 5 micromachines-14-02217-f005:**
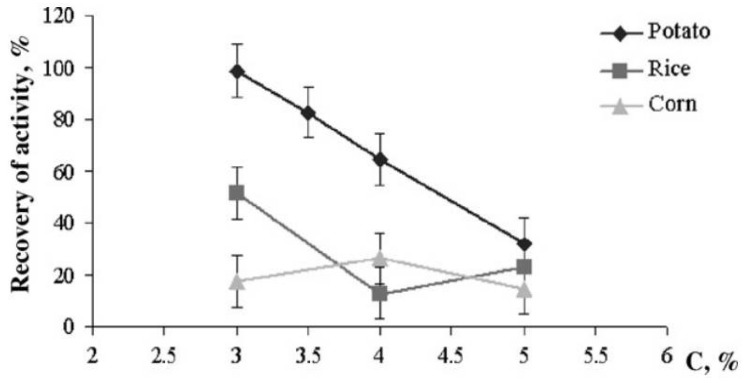
The recovery of activity of the bi-enzymatic Red + Luc system immobilized into potato, corn, and rice starch gels with different starch concentrations. The reaction mixture includes disks with immobilized enzymes and solutions of all needed substrates (tetradecanal, FMN, and NADH). Reproduced from ref. [71] with permission from Elsevier Inc.

**Figure 6 micromachines-14-02217-f006:**
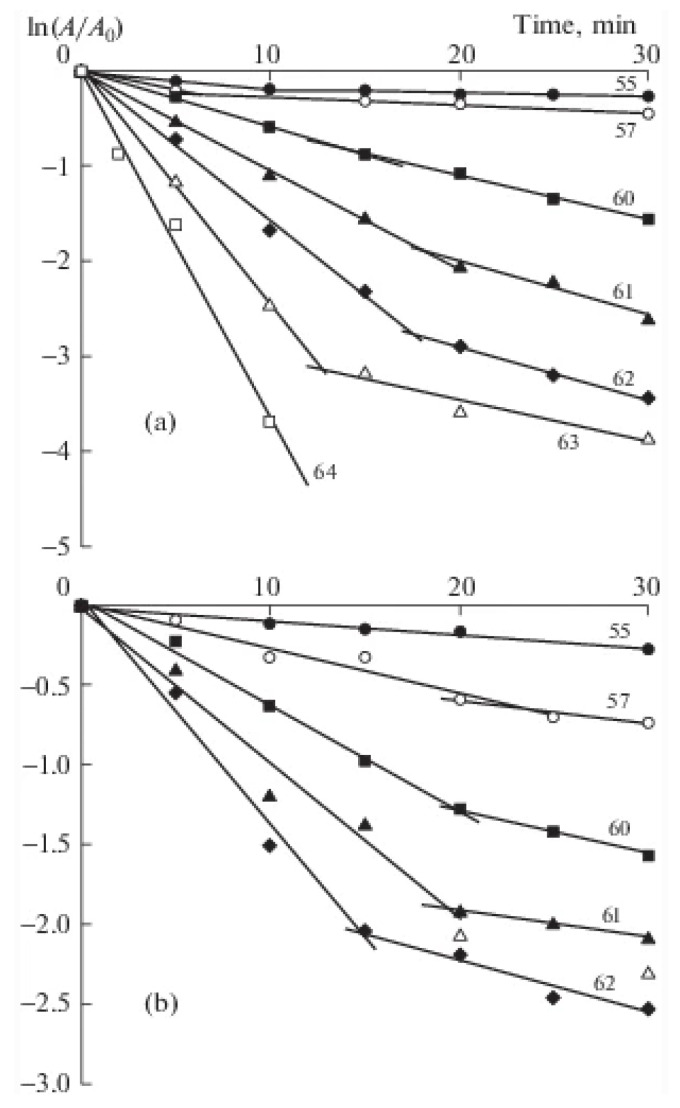
Kinetic curves of the thermal inactivation of BChE in (**a**) a 1.4% gelatin solution and (**b**) a 3% starch solution at various temperatures. Reproduced from ref. [101] with permission from Pleiades Publishing.

**Figure 7 micromachines-14-02217-f007:**
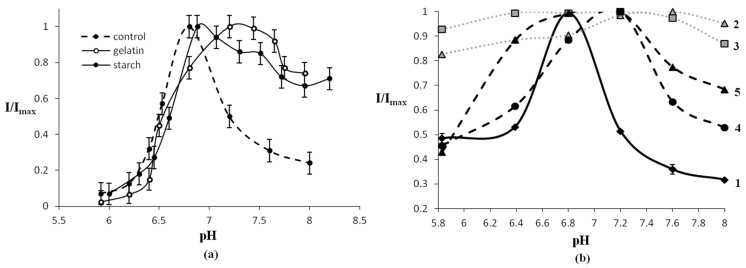
pH dependence of the normalized luminescence intensity of Red + Luc: (**a**) in buffer solution (control) and in the presence of 1% gelatin or 2% starch gels; (**b**) in immobilized enzyme preparations in the form of *dried* disks: (1) free Red + Luc; (2) Red + Luc immobilized in starch gel; (3) Red + Luc +C_14_+NADH immobilized in starch gel; (4) Red + Luc immobilized in gelatin gel; (5) Red + Luc +C_14_+NADH immobilized in gelatin gel. Reproduced from refs. [97,102] with permission from Pleiades Publishing.

**Table 1 micromachines-14-02217-t001:** Immobilization yield of enzymes entrapped in the starch and gelatin gels.

Enzymes (Content)	Conditions	Immobilization Yield, %		Reference
	Starch/Gelatin	Starch	Gelatin	
Red + Luc (0.1 mU of Red and 0.2 μg of Luc)	3% potato starch gel, 4% gelatin gel	100	17	[71]
FLuc (5.5 × 10^12^ RU·mg^−1^)	1% of gelatin, 0.5 mM DTT	- *	60	[59]
BChE (60 mU)	3% potato starch gel, 1.4% gelatin gel	90	100	[58,85]
Try (11 U)	3% potato starch gel	100	- **	[60]

*—Starch gel was unsuitable for immobilization of FLuc. **—No Try immobilization into gelatin gel was performed.

**Table 2 micromachines-14-02217-t002:** Parameters of the Red + Luc bi-enzymatic system of luminous bacteria (0.1 mU of Red and 0.2 μg of Luc).

Enzyme Forms	Km App ^#^, mg/mL			pH Optimum *	T,°C Optimum *
	NADH	FMN	Tetradecanal		
Free	(3.4 ± 0.7) × 10^−2^	(0.6 ± 0.1) × 10^−3^	(0.18 ± 0.04) × 10^−6^	6.7–6.9	20–25
Immobilized in potato starch gel	(9.0 ± 1.8) × 10^−2^	(1.3 ± 0.3) × 10^−3^	(1.2 ± 0.2) × 10^−6^	5.8–8.0	5–50
Immobilized in gelatin gel	(17.9 ± 3.5) × 10^−2^	(2.8 ± 0.6) × 10^−3^	(0.7 ± 0.1) × 10^−6^	6.6–7.3	15–50

^#^–Apparent Michaelis constants (Km app) were determined by analyzing the relationships between the reaction rate and concentrations of substrates in the Eadie–Hofstee coordinates. *–the range of the values of the factor (pH or T) at which the enzymes lose no more than 10% of their activity.

## Data Availability

The data presented in this study are available in the article.

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
