# Peer review of "Enzymes Immobilized into Starch- and Gelatin-Based Hydrogels: Properties and Application in Inhibition Assay"

_micromachines, 2023, doi:10.3390/mi14122217_

Round 1

Reviewer 1 Report

Comments and Suggestions for Authors

The review article “Enzymes Immobilized into Starch- and Gelatin-Based Hydrogels: Properties and Application in Inhibition Assay” is devoted to a modern and relevant topic - promising carriers for enzyme immobilization, namely hydrogels based on natural biodegradable polymers, starch and gelatin.

However, the review needs serious improvement.

1. The review contains 20 works of the co-authors of this article in References list, which is about 23% of all references. I don't think this is good.

2. Only 23 references (about 27% of all references) are dated 2019-2023, which is quite small for good review works. Authors should increase the proportion of references dated within the last 5 years.

3. Lines 51-56: the authors provide information about agarose and guar gum, chitin and chitosan, alginate hydrogels as carriers for enzyme immobilization. It is not clear why this paragraph is in the review if the article which is devoted to starch- and gelatin-based hydrogels.

4. Table. 1 and Section 3.3 include references only from the co-authors of this review. It turns out that, apart from them, no one had previously counted on immobilization yield of enzymes entrapped in the starch and gelatin gels and did not deal with the problems of the effect of starch and gelatin gels matrix on stability of the immobilized enzymes. In addition to references to the works of other authors in Table. 1 it would be good to add information on temperature and pH optima, Km and Vmax for immobilized enzymes.

5. The conclusion is written very sparsely; trends in the development of science in the field of starch- and gelatin-based hydrogels as matrix for enzymes immobilization, problems and prospects for working with these types of carriers should be added to it.

Technical Notes:

Line 89: probably a typo - instead of “amylase” it should be “amylose”.

Lines 216-225: the font in this paragraph is smaller than in the main text of the article.

Comments on the Quality of English Language

Minor editing of English language required

Author Response

Dear Reviewer,

Thank you for your careful consideration and valuable comments about our article. We were pleased to answer your questions. 

Point 1: The review contains 20 works of the co-authors of this article in References list, which is about 23% of all references. I don't think this is good.

Response 1:  Thank you for the comment. We have added 23 new references by other authors.  The main goal of the review was to summarize our long-term experience in using starch and gelatin gels without additional additives and modifications to obtain immobilized enzyme preparations. Therefore, the number of references to our own publications remains quite significant.

Point 2:  Only 23 references (about 27% of all references) are dated 2019-2023, which is quite small for good review works. Authors should increase the proportion of references dated within the last 5 years.

Response 2: We have significantly increased the proportion of references dated within the last 5 years. Currently, the number of references dated 2019-2023 is 43 (about 39% of all references).

Point 3: Lines 51-56: the authors provide information about agarose and guar gum, chitin and chitosan, alginate hydrogels as carriers for enzyme immobilization. It is not clear why this paragraph is in the review if the article which is devoted to starch- and gelatin-based hydrogels.

Response 3:  We mention agarose, guar gum, chitin, chitosan and alginate only in the introductory part as an examples of other natural carriers for enzymes. We believe it is relevant because recently they are the most widely used natural biodegradable polymer materials suitable for enzyme immobilization.

Point 4: Table. 1 and Section 3.3 include references only from the co-authors of this review. It turns out that, apart from them, no one had previously counted on immobilization yield of enzymes entrapped in the starch and gelatin gels and did not deal with the problems of the effect of starch and gelatin gels matrix on stability of the immobilized enzymes. In addition to references to the works of other authors in Table. 1 it would be good to add information on temperature and pH optima, Km and Vmax for immobilized enzymes.

 Response 4: In fact, there are very few published data on the immobilization yield of enzymes entrapped into starch and gelatin gels, since these polymers are very little used for enzyme immobilization in their native form (without any additives and modifications). In articles on drag delivery systems, entrapment into gelatin hydrogels, even unmodified ones, can be found, but the immobilization yield of enzymes in such cases is not measured. To give some example and comparison, we added the value of the immobilization yield of bovine liver catalase during immobilization in gelatin-alginate hydrogel.

Page 9, lines 348-351: “Although other authors also reported successful immobilization of enzymes in gelatin gel, an essential difference was the use of additional components, primarily alginate. The activity yield of bovine liver catalase immobilized in gelatin-alginate hydrogel was 92% [68].”

  1. Abdel-Mageed HM, Abd El Aziz AE, Abdel Raouf BM, Mohamed SA, Nada D. Antioxidant-biocompatible and stable catalase-based gelatin-alginate hydrogel scaffold with thermal wound healing capability: immobilization and delivery approach. 3 Biotech. 2022, 12, 73. https://doi.org/10.1007/s13205-022-03131-4

In addition, as an example, comparative information on the optimal pH and temperature, as well as Km values, for the immobilized bi-enzymatic system of luminous bacteria Red + Luc was added to the Section 3.3 (Table 2). This system is the most studied in terms of the effects of starch and gelatin hydrogels on enzymatic kinetics and stability under various physical and chemical factors.

Pages 10-11, Lines 417-438: “The most extensive data have been obtained on the effects of starch and gelatin hy-drogels on the kinetic properties of the bi-enzymatic system of luminous bacteria Red + Luc. The study [98] demonstrated an increase in the apparent Michaelis constants (Km app) in the immobilized Red + Luc systems for all three substrates compared with the free enzymes (Table 2), which can be explained by limitation of the conformational lability of protein molecules reflected in the rate of the catalytic activity. Another reason for the in-crease in the Michaelis constants may be complications with the delivery of the substrates to the enzymes immobilized in the hydrogels [98].

The reason for the diffusion limitations, which are the common drawback of immobilized enzymes, is that for an effective enzymatic catalysis, the substrates should reach the active site of the immobilized enzyme through the reaction medium and the products should diffuse away from it. To overcome this drawback, a number of strategies are employed, including thermally reversible hydrogels and carriers, the immobilization of enzymes on the surface of the carrier, and others [99]. Co-immobilization of the coupled enzymes and their substrate is also a helpful measure to minimize diffusion limitations.”

  1. Esimbekova, E.N.; Torgashina, I.G.; Kratasyuk, V.A. Comparative Study of Immobilized and Soluble NADH:FMN-oxidoreductase–luciferase coupled enzyme system. Biochem (Mosc) 2009, 74, 695-700. https://doi.org/10.1134/s0006297909060157
  2. Malar, C.G.; Seenuvasan, M.; Kumar, K.S.; Kumar, A;, Parthiban, R. Review on surface modification of nanocarriers to overcome diffusion limitations: An enzyme immobilization aspect. Biochemical engineering journal, 2020, 158, 107574. https://doi.org/10.1016/j.bej.2020.107574.

Point 5: The conclusion is written very sparsely; trends in the development of science in the field of starch- and gelatin-based hydrogels as matrix for enzymes immobilization, problems and prospects for working with these types of carriers should be added to it.

Response 5: Thank you for the comment. We have added to the Conclusion information about the existing problems of using starch and gelatin-based hydrogels as a matrix for enzyme immobilization, and indicated how researchers can solve them.

Page 15, Lines 585-591: “Despite the many advantages of using natural hydrogels to immobilize enzymes, certain problems arise. For example, retrogradation or destruction of native starch changes the physico-chemical properties of the gels, which may lead to changes in the properties of the enzymes immobilized in the starch gel. To avoid this, researchers suggest performing additional modifications of starch and gelatin, among which crosslinking is the most widely used procedure. Another likely problem is diffusion limitations. This problem can be sometimes overcome by conducting multi-enzyme immobilization.”

In addition, a typo has been corrected: "amylase" has been corrected to "amylose". The font in the text is aligned with the rules of the journal.

Reviewer 2 Report

Comments and Suggestions for Authors

This review article describes the current state of immobilizing enzymes in hydrogels, specifically biodegradable polymers such as starch and gelatin.  Overall the authors do a thorough job in describing the pluses and minuses of both biopolymers for enzyme immobilization.  The also discuss some of the uses of enzyme encapsulated into these biopolymers.  

Overall, this is a nice review and I suggest it be accepted with only minor editing, mostly due to grammatical mistakes.

Comments on the Quality of English Language

For the most part, the English is excellent.  I found about a dozen places where the authors left out articles such as a, an, the that are typically used before nouns.  A careful editing by a native English speaker is warranted.

Author Response

Dear Reviewer,

Thank you for your comments and suggestions. We greatly appreciate your kind words.

The English language has been corrected.

Round 2

Reviewer 1 Report

Comments and Suggestions for Authors

Authors corrected the manuscript according to my comments